# Tissue-specific (ts)CRISPR as an efficient strategy for in vivo screening in *Drosophila*

Hagar Meltzer[1], Efrat Marom[1], Idan Alyagor[1], Oded Mayseless[1], Victoria Berkun[1], Netta Segal-Gilboa[2], Tamar Unger[2], David Luginbuhl[3] & Oren Schuldiner [1]

Gene editing by CRISPR/Cas9 is commonly used to generate germline mutations or perform in vitro screens, but applicability for in vivo screening has so far been limited. Recently, it was shown that in *Drosophila*, Cas9 expression could be limited to a desired group of cells, allowing tissue-specific mutagenesis. Here, we thoroughly characterize tissue-specific (ts) CRISPR within the complex neuronal system of the *Drosophila* mushroom body. We report the generation of a library of gRNA-expressing plasmids and fly lines using optimized tools, which provides a valuable resource to the fly community. We demonstrate the application of our library in a large-scale in vivo screen, which reveals insights into developmental neuronal remodeling.

[1] Department of Molecular Cell Biology, Weizmann Institute of Science, Rehovot, Israel. [2] Structural Proteomics Unit, Weizmann Institute of Science, Rehovot, Israel. [3] Howard Hughes Medical Institute, Department of Biology, Stanford University, Stanford, USA. Correspondence and requests for materials should be addressed to O.S. (email: oren.schuldiner@weizmann.ac.il)

Over the past few years, clustered regularly interspaced short palindromic repeats (CRISPR)/CRISPR-associated protein (Cas) has been demonstrated as an efficient genome-editing tool for the study of diverse biological questions in countless organisms and cell types[1,2]. Among its many applications, CRISPR has proved to be a useful strategy for conducting high-throughput screens[3,4], an important method for uncovering novel genes involved in complex biological processes. To date, most CRISPR-mediated screens were conducted in vitro within cell cultures[5–7]. Several studies also reported transplantation of CRISPR-targeted cells into living organisms for screening in an in vivo context[8,9]. However, reports of direct in vivo screens—in which CRISPR is used to mutate the endogenous animal tissue—are scarce, as they require complex endeavors that limit the scope of the screen and its efficiency[10,11].

In *Drosophila melanogaster*, CRISPR is routinely used for the generation of heritable germline mutations, including small insertions/deletions (indels) or deletion of large DNA fragments, by driving transgenic Cas9 expression using a promoter that is active in the germline (such as the *nos* promoter)[12,13]. Recently, it was demonstrated that CRISPR could also be successfully applied in *Drosophila* in a tissue-specific manner (hereby referred to as tsCRISPR), which restricts mutagenesis to a desired somatic tissue or group of cells. This is most commonly achieved using the binary GAL4/UAS system[14–16], which allows the fly to express Cas9 in any tissue of interest. Alternatively, intermediate GAL4 production can be bypassed by using enhancer-fusion constructs to allow Cas9 expression[17,18]. Combined with transgenic expression of a guide-RNA (gRNA) targeting the gene of interest, this results in tissue-specific biallelic gene disruption within a WT environment (Fig. 1a). The simplicity and modular nature of tsCRISPR make it ideal for high-throughput in vivo screening.

While one of the major strengths of *Drosophila* as a model organism is its high suitability for genetic screens, common screening methods hold various limitations. Strategies based on mutation analysis often require the application of labor-intensive and time-consuming mosaic techniques, due to the high rate of homozygous lethal mutations. Alternatively, RNA-interference (RNAi)-based screens suffer from partial gene knockdown[19]. tsCRISPR has the potential to overcome the limitations of current screening methods while enjoying their advantages: like RNAi, it is a rapid process which only requires a single cross, however unlike RNAi, disruption occurs at the DNA level to achieve complete genetic knockout. Despite its potential, to date, tsCRISPR was not reported in a large-scale loss-of-function screen. This is, in part, due to the need for further characterization of this state-of-the-art technique and its growing body of reagents.

Neuronal remodeling is a conserved late-developmental mechanism to refine neural circuits, which often combines both degenerative and regenerative events[20]. Defects in remodeling have been associated with neurologic disorders such as schizophrenia and Alzheimer's disease[21,22]. In the *Drosophila* brain, mushroom body (MB) γ neurons undergo remodeling in a highly stereotypical manner during metamorphosis, including pruning of larval axons followed by regrowth of adult-specific ones[23] (Fig. 1b). Despite the recent progress in identifying the genes and pathways involved in MB remodeling[23,24], much of the molecular basis underlying this process remains unknown. Recently, we uncovered the transcriptional landscape of developing γ neurons at fine temporal resolution[25]. This highlighted many candidate genes whose unique expression implies potential involvement in remodeling.

In this paper, we first characterize various tsCRISPR tools and optimize their use in a complex neuronal system. Next, we report the generation of a resource of flies and plasmids harboring gRNAs for specific genes, that could be highly beneficial for the entire fly community. Finally, and most importantly, we demonstrate the application of the tsCRISPR strategy in a large-scale in vivo screen aimed to uncover molecules required for developmental neuronal remodeling.

## Results

**tsCRISPR is efficient in the fly central nervous system**. To establish the feasibility of tsCRISPR as an efficient screening strategy in MB γ neurons, we performed a proof-of-concept study using nine genes with known and published roles in MB γ axon pruning (Supplementary Table 1). For each gene, three different gRNA sequences were cloned into the pCFD3 plasmid, which allows ubiquitous expression of a single gRNA from the U6:3 promoter[14]. Each gRNA-expressing plasmid was used to generate a transgenic fly line, and the 27 lines were crossed to a line that expresses Cas9 specifically in MB γ neurons, using the γ-specific GMR71G10-GAL4[25]. While the observed pruning defects were highly similar to those of neuroblast mosaic analysis with a repressible cell marker (MARCM)[26] clones homozygous for mutations derived from the germline, RNAi lines targeting the same genes yielded much weaker, and more variable, phenotypic effects (Fig. 1c, Supplementary Fig. 1a, Supplementary Table 1). The vast majority (81%) of gRNA lines induced a phenotype that was detectable in at least 50% of the brains, while only about a fifth seemed to be ineffective. In contrast, available RNAi lines targeting the same pruning-related genes displayed dramatically reduced phenotypic penetrance, with more than half of the lines showing a WT phenotype, and only a minority (13%) demonstrating phenotypic penetrance of over 50% (Fig. 1d), validating tsCRISPR as a more consistent and efficient targeting tool than RNAi in our experimental system. A broad analysis of the gRNAs revealed, as previously reported[27], that high GC content in the protospacer adjacent motif (PAM)-adjacent region of the gRNA sequence was correlated with significantly increased efficiency (Fig. 1e). To better characterize gene disruption efficiency, we stained for proteins encoded by tsCRISPR-targeted genes, including *EcR*[28] and *Bsk*[29]. When using GMR71G10-GAL4 to drive Cas9 expression, we detected a dramatic decrease in the proportion of immunoreactive MB γ cell bodies or axons, indicating a high rate of protein null mutations (Supplementary Fig. 1b–c; also see reduction in Mamo in next section). We observed a similar decrease in immunoreactivity in glial cell bodies in the ventral nerve cord when using the glia-specific Repo-GAL4 (Supplementary Fig. 1d). Taken together, these results suggest that tsCRISPR is an efficient method for gene disruption in the fly central nervous system, and that compared to RNAi, it is expected to constitute a more efficient screening strategy and yield fewer false-negative results.

**A thorough comparison of different gRNA-expressing vectors**. Since we found that about one-fifth of tested gRNAs were ineffective, we decided to express two gRNAs targeting each gene, and therefore tested the in vivo efficiency of various pCFD plasmids[14,15]. pCFD4 enables expression of two different gRNA sequences under two different U6 promoters, U6:1 and U6:3, the latter shown to induce stronger activity[14]. In pCFD5, both gRNAs are transcribed as a single transcript from the stronger U6:3 promoter, and later excised by the endogenous cell machinery owing to flanking tRNAs[15]. We observed prominent pruning defects when driving ubiquitous expression of two *Plum*-gRNAs using either pCFD4 or pCFD5, the latter being significantly more severe (linear mixed effects model: $p < 0.001$, Fig. 2a, ranked independently by two investigators in Fig. 2c and Supplementary Fig. 2a). In parallel, we tested protein reduction by driving

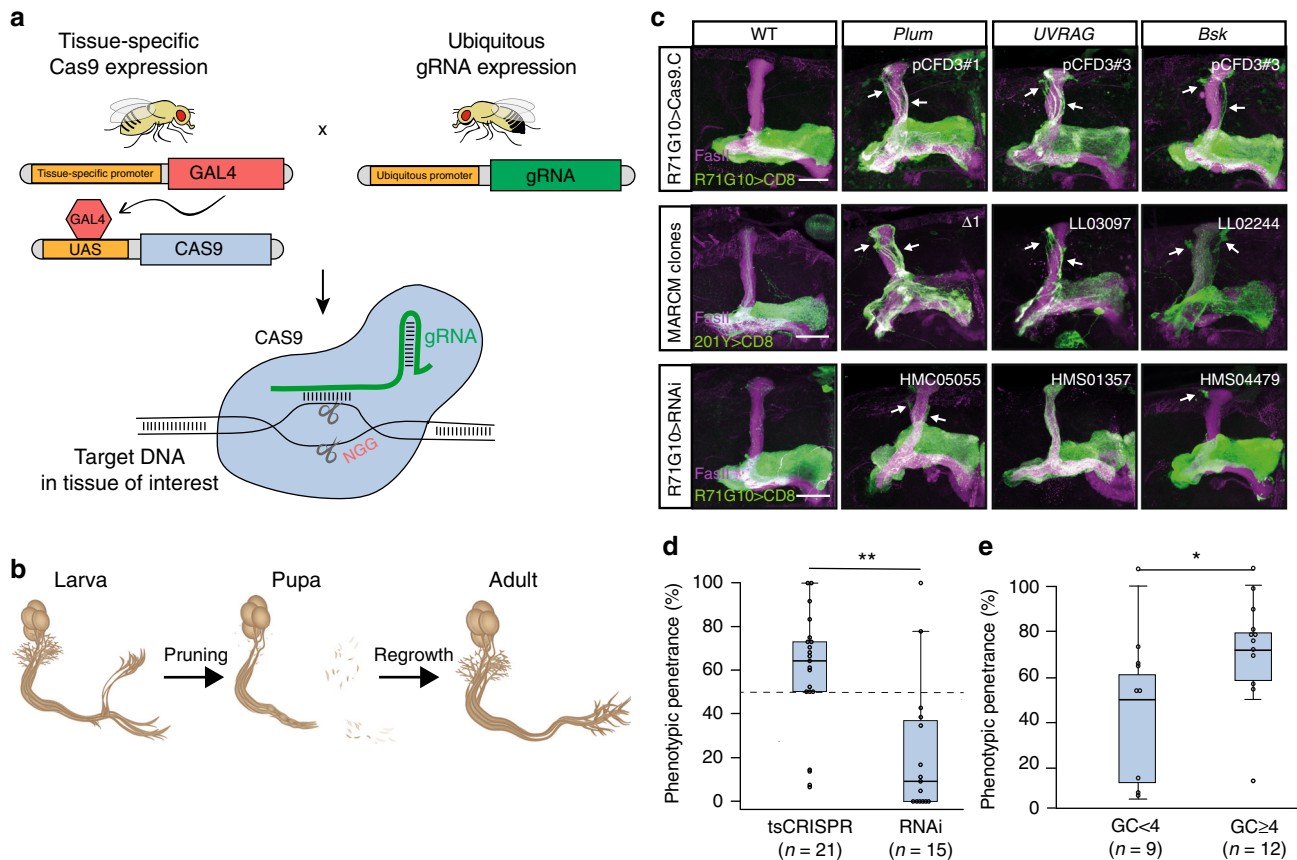

**Fig. 1** tsCRISPR is efficient in the fly central nervous system. **a** Schematic representation of tsCRISPR using the GAL4/UAS system. **b** Scheme of MB γ neuronal remodeling throughout metamorphosis. (Adapted from ref. [24]). **c** Confocal Z-projections of adult MBs expressing the indicated gRNAs as well as *UAS-Cas9.C* and membrane bound CD8::GFP (CD8) driven by *R71G10-GAL4* (upper row); or MB neuroblast MARCM clones of the indicated alleles labeled by CD8 driven by *201Y-GAL4* (middle row); or MBs expressing the indicated TRiP RNAi's and CD8 driven by *R71G10-GAL4* (lower row). While *R71G10-GAL4* is γ neuron-specific, *201Y-GAL4* is also expressed in a subset of the later born α/β neurons that project dorsally as a tight fascicle. **d** Phenotypic penetrance of gRNA and RNAi lines targeting the same 9 pruning-related genes. Mann–Whitney $U$ test: $W = 258$, **$p = 0.0013$. **e** Phenotypic penetrance of gRNA lines divided into two categories of GC-content in the PAM-adjacent region ($\geq$ or <than 4 GC's within the 6 nucleotides at the 3' end the gRNA sequence). Mann–Whitney $U$ test: $W = 25.5$, *$p = 0.0462$. Source data are provided as a Source Data file

*Mamo*-gRNAs, in which case pCFD4 and pCFD5 did not differ significantly, but pCFD5 was more consistent (Fig. 2b, d). The use of pCFD3 (which allows ubiquitous expression of a single gRNA) resulted in significantly reduced severity of the observed pruning phenotype (linear mixed effects model: $p = 0.001$ and $p < 0.001$ compared to pCFD4 and pCFD5, respectively, Fig. 2c and Supplementary Fig. 2a), and less efficient reduction of protein staining (Fig. 2b, d). This efficiency ranking of pCDF5 > pCDF4 > pCDF3 is consistent with previous findings[14,15]. Interestingly, for pCFD6, in which the U6 promoter of pCFD5 was replaced by UAS to allow tissue-specific rather than ubiquitous gRNA expression[15], the *plum* pruning defect was similar in severity to pCFD4 but significantly weaker than pCFD5 (linear mixed effects model: $p < 0.001$, Fig. 2a, c, Supplementary Fig. 2a), however, reduction in *mamo* protein staining was significantly less efficient than both pCFD4 and pCFD5 (Fig. 2b, d). This suggests that pCFD6 efficiency might be gene or GAL4-specific. Altogether, our results demonstrate that pCFD5 is superior to the other pCFD plasmids in terms of targeting efficiency.

**Cas9.P2 is less potent than Cas9.C and may reduce lethality.** One concern in the field is the potential lethality associated with tsCRISPR. It has been reported that high levels of Cas9 may be cytotoxic and induce lethality even in the absence of gRNAs,

especially when driven by strong GAL4s[14,17,18], yet we have not observed toxicity with our selected driver GMR71G10-GAL4. We did, however, encounter 6 gRNA lines in our proof-of-concept study (targeting two different genes—*EcR* and *Mov34*[30]) for which tsCRISPR was lethal. Crossing these gRNA-lines with *UAS-Cas9.C* remained lethal even in the absence of GAL4. This indicates leaky expression of Cas9 from the UAS promoter (as previously reported[15]), which may be lethal when causing ectopic mutagenesis in a tissue or organ in which the targeted gene is vital (specifically, the lethality associated with *EcR*-gRNAs suggests that Cas9.C is expressed in a leaky manner in tissues critical for metamorphosis). To try to resolve this, we used another variant of UAS-Cas9, *UAS-Cas9.P2*, designed to be expressed in lower levels[15]. For reasons that remain unknown, this only rescued lethality in two of the gRNA lines (one for each gene), which now displayed the expected pruning defect phenotype (although milder than expected in the case of *Mov34*, Supplementary Fig. 3a). For the remaining lines, tsCRISPR using *UAS-Cas9.P2* was still lethal, however this lethality was now GAL4-dependent, and did not occur in its absence. This suggests that lethality is gene-specific and might be a combination of leaky expression of Cas9, as in the case of *UAS-Cas9.C*, or GAL4-dependent (presumably in non-MB tissues) vital gene requirement, as in the case of *Cas9.P2*. Despite its apparent advantage in reducing lethal crosses, we observed significantly decreased efficiency when comparing the

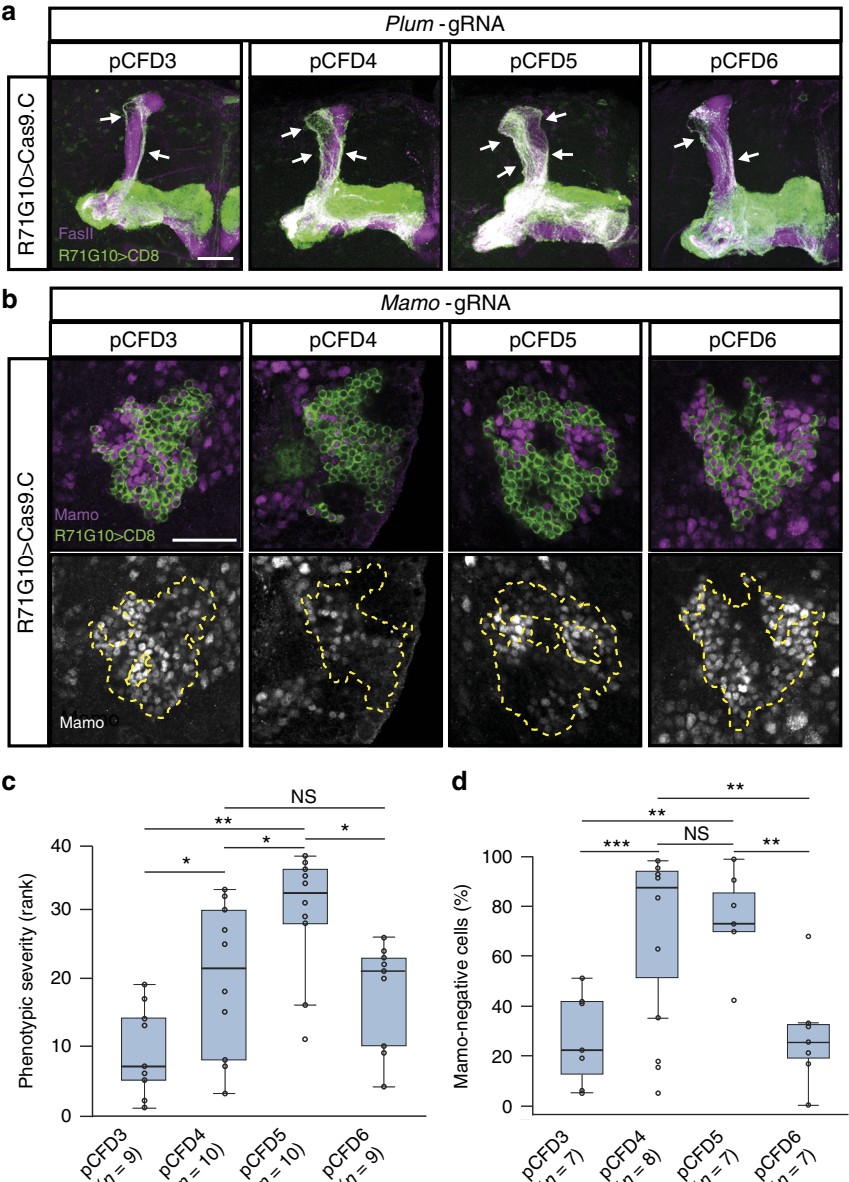

**Fig. 2** A thorough comparison of different gRNA-expressing vectors. **a** Confocal Z-projections of adult MBs expressing *Plum*-gRNAs using pCFD3/4/5/6, as well as *UAS-Cas9.C* and CD8 driven by *R71G10-GAL4*. **b** Single-confocal sections of the MB cell body region 6 h after puparium formation (APF), expressing *Mamo*-gRNAs using pCFD3/4/5/6. *R71G10-GAL4* drives expression of *UAS-Cas9.C* and CD8, and cell bodies are stained for Mamo (magenta or gray). **c** Quantification of (a): Kruskal–Wallis test: $\chi^2_{(3)} = 15.92$, $p = 0.0012$; Pairwise Wilcoxon test (FDR correction): pCFD3–pCFD4: *$p = 0.04$; pCFD4–pCFD5: *$p = 0.04$; pCFD3–pCFD5: **$p = 0.004$; pCFD5–pCFD6: *$p = 0.017$. For simplicity, the plot only displays results from one ranker, and the results of the second ranker are presented in Supplementary Fig. 2. **d** Quantification of (b): one-way anova: $F_{(3,25)} = 12.2$, $p < 0.001$; Tukey's test: pCFD3–pCFD4: ***$p < 0.001$; pCFD3–pCFD5: **$p = 0.001$; pCFD5–pCFD6: **$p = 0.002$; pCFD4–pCFD6: **$p = 0.001$. In all confocal images, scale bar represents 30 μm. White arrows highlight unpruned γ axons. Yellow dashed lines demarcate the Cas9 expression domain. Unless otherwise stated, CD8, green; FasII, magenta (strongly stains α/β and weakly stains γ neurons). In all boxplots, the box represents first to third quartiles, whiskers represent minimum and maximum values that are within 1.5× interquartile range, horizontal line represents the median, and empty circles represent all values within the group. Source data are provided as a Source Data file

*Cas9.P2* variant to the stronger *Cas9.C* (both phenotypically and by staining for proteins encoded by tsCRISPR-targeted genes, Supplementary Fig. 3b–d). We therefore decided that its use is less advisable in the context of large-scale screening, keeping in mind that we are looking for abnormal phenotypes in the MB and are less concerned by ectopic mutagenesis. Importantly, during the screen (see later), lethality was observed in only about 1% of the crosses (despite the use of *UAS-Cas9.C*), indicating that at least in our particular screening setup, *Cas9.C* leakiness-induced lethality is practically

negligible. We acknowledge, however, that in other screening contexts, specifically when a large proportion of the screened genes are essential, *Cas9.C*-associated lethality might be more common, and therefore the decision which Cas9 to use should be made individually for each screen. One possible strategy is to initially screen with *Cas9.C*, but repeat all lethal crosses with *Cas9.P2*.

**tsCRISPR screen reveals unknown neuronal remodeling genes**. Once we established tsCRISPR as a highly efficient tool for

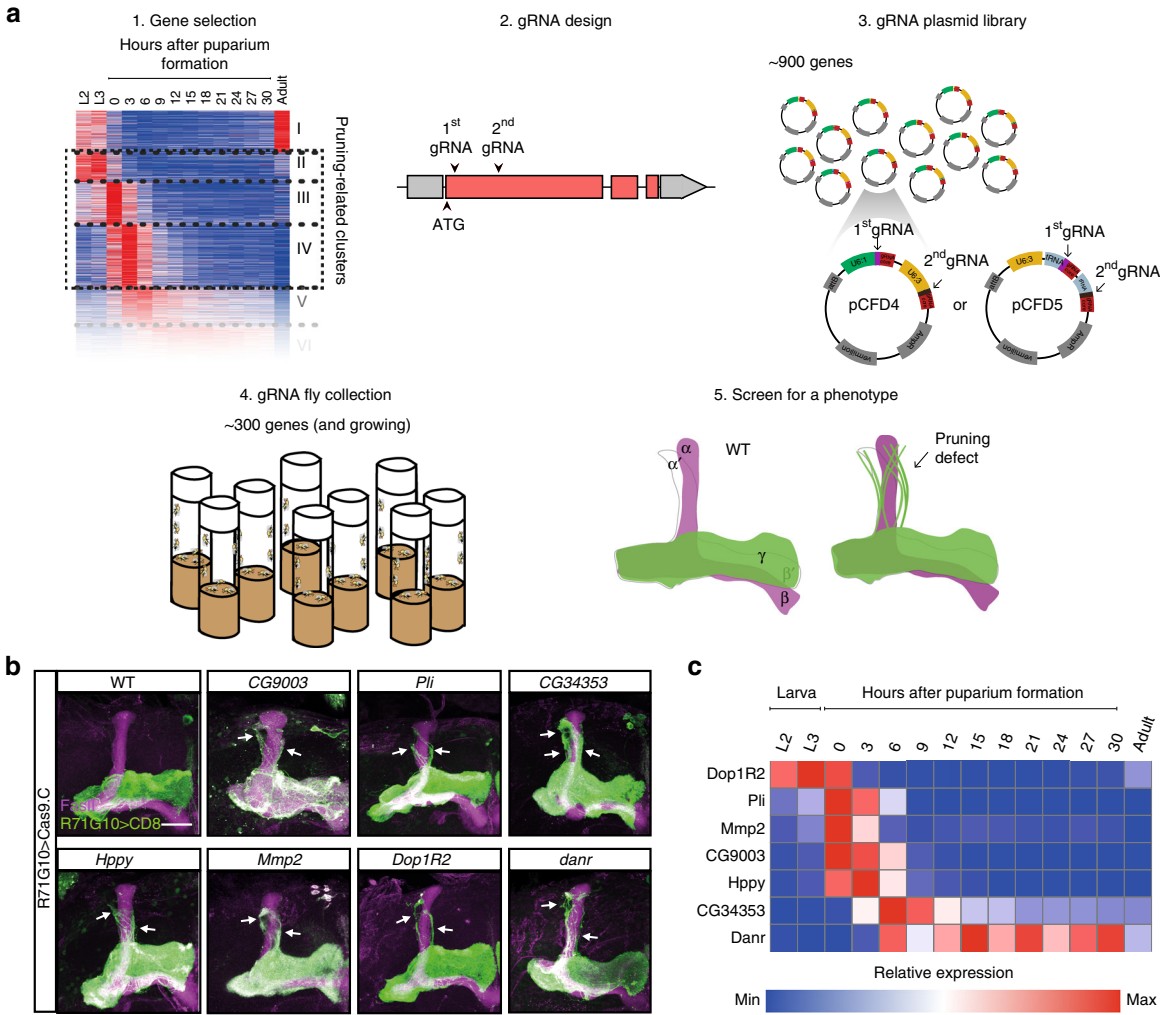

**Fig. 3** tsCRISPR screen reveals unknown neuronal remodeling genes. **a** Outline of the screen, including the generation of the gRNA fly collection (items 1,5 adapted from ref. [25]). **b** Confocal Z-projections of adult MBs expressing gRNAs targeting the indicated genes (in pCFD4), as well as *UAS-Cas9.C* and CD8 driven by *R71G10-GAL4*. White arrows highlight unpruned γ axons. **c** Heat-map displaying the expression profiles of the genes in **b** within MB γ neurons throughout development[25]. Red and blue represent high- and low-relative expression of each gene during development, respectively. Source data are provided as a Source Data file

biallelic gene disruption in MB γ neurons, our next step was to apply it in a large-scale screen in search of genes involved in axon pruning (Fig. 3a). To this end, we generated a library of gRNA-expressing pCFD4 or pCFD5 constructs (shifting from pCFD4 to pCFD5 as we progressed). Our library currently holds approximately 900 plasmids, each harboring two different gRNAs targeting each *Drosophila* gene of interest (Supplementary Data 1). The rapid and high-throughput cloning strategies of two gRNAs into pCFD4 and pCFD5 are detailed in the methods section. While the majority of genes were selected based on our MB γ neuron developmental expression data, we also included genes that encompass broader interest, for example neurotransmitter receptors and key players in developmental signaling pathways. To maximize the likelihood of a cleavage event that would lead to an indel mutation resulting in an early frame-shift and strong loss-of-function, for each gene we aimed to choose two gRNA sequences which follow the guideline of a GC-rich PAM-adjacent region, and which are both located within the coding region immediately downstream of the translation initiation site. So far, we used our plasmid library to generate a fly resource of approximately 300 confirmed transgenic gRNA-expressing *Drosophila* lines. Our constantly growing gRNA fly collection poses

an extremely valuable resource, available to the worldwide fly community (transgenic flies available in the Bloomington Drosophila Stock Center and plasmids available in Drosophila Genomics Resource Center; see further details in the methods section). Flies harboring gRNAs can be readily used for large-scale screening in any desired tissue, or alternatively for the rapid generation of germline mutations in specific genes of interest.

Our ongoing screen has so far identified several genes, involving various molecular pathways, for which tsCRISPR yielded γ axon pruning defects (Fig. 3b). The expression patterns of all genes[25] match their potential, previously unknown roles in MB γ axon pruning (Fig. 3c), and all make for interesting candidates for further study that might shed light on the process of neuronal remodeling.

**The F-BOX protein CG9003 is important for MB pruning.** Among the genes we identified is *CG9003*, which, based on its sequence, encodes an F-BOX protein predicted to function in the Skp1-Cullin-F-BOX (SCF) E3 ligase complex. Within the SCF complex, F-BOX proteins determine substrate specificity[31,32]. To further validate the pruning defect induced by tsCRISPR, we

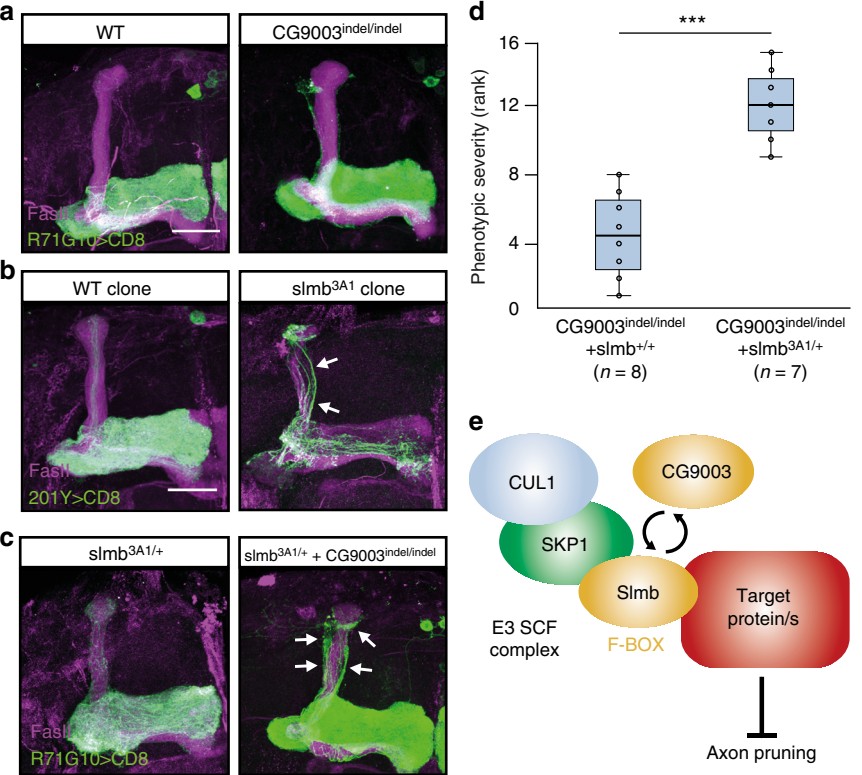

**Fig. 4** The F-BOX protein CG9003 is important for MB pruning. **a** Confocal Z-projections of MBs of adult flies that are either WT or homozygous for *CG9003*[indel]. CD8 is driven by *R71G10-GAL4*. **b** Confocal Z-projections of WT or *slmb3A1* MB neuroblast MARCM clones, labeled by CD8 driven by *201Y-GAL4*. **c** Confocal Z-projections of MBs of adult flies that are heterozygous for the *slmb3A1* mutation, and in addition either WT in the *CG9003* gene (left) or homozygous for the *CG9003*[indel] mutation (right). CD8 is driven by *R71G10-GAL4*. **d** Boxplot depicting the rank of pruning defect severity of *CG9003*[indel] homozygotes without or with *slmb3A1* heterozygosity. The box represents the first to third quartiles, whiskers represent minimum and maximum values within 1.5× interquartile range, the horizontal line represents the median, and empty circles represent all values within the group. Mann–Whitney *U* test: $W = 0$, ***$p < 0.001$. For simplicity, the plot only displays results from one ranker, and the results of the second ranker are presented in Supplementary Fig. 4d. **e** Schematic suggested model of the function of F-BOX proteins CG9003 and Slmb in MB γ axon pruning. Either Slmb or CG9003 can function as the F-BOX protein of the SCF complex, and either of them can bind a shared substrate whose degradation is essential for proper execution of pruning. In all confocal images, scale bar represents 30 μm. White arrows highlight unpruned γ axons. CD8, green; FasII, magenta. Source data are provided as a Source Data file

expressed an RNAi line targeting CG9003 that also displayed a mild pruning defect (Supplementary Fig. 4a), and we also used our existing gRNA line to generate a germline mutation. This extremely simple transition, which merely requires crossing the gRNA line to a Cas9 that is expressed in germ cells, resulted in an indel mutation (*CG9003*[indel]), expected to encode a truncated protein that lacks the F-BOX domain (as well as all other predicted domains, Supplementary Fig. 4b). While MB γ axons of flies homozygous for *CG9003*[indel] grow normally (Supplementary Fig. 4c), adult MBs consistently display pruning defects (Fig. 4a). It was previously reported that *Supernumerary limbs* (*Slmb*), another *Drosophila* F-BOX protein, is required for pruning of both dendritic arborization (da) and MB γ neurons[31]. Indeed, homozygous mutant clones for the existing *slmb3A1* allele[33] demonstrated a severe pruning defect (Fig. 4b). While *slmb3A1* heterozygote flies had WT MBs, combining heterozygous *slmb3A1* with homozygous *CG9003*[indel] resulted in a pruning defect that was significantly more severe than that of homozygous *CG9003*[indel] alone (linear mixed effects model: $p < 0.001$, Fig. 4c, two independent ranking evaluations shown in Fig. 4d and Supplementary Fig. 4d). This suggests that the F-BOX proteins *Slmb* and *CG9003* share at least one downstream target, whose degradation is essential for proper progression of MB γ axon pruning (Fig. 4e). While in da neurons the target of *Slmb* is reported to be the Tor pathway, in MB γ neurons this pathway

does not seem to play a role in pruning[31,34]. Therefore, the identity of the common target(s) of *CG9003* and *Slmb* remains to be identified.

## Discussion

In this paper, we conducted a thorough characterization of tsCRISPR in a complex neuronal system, followed by a large-scale (ongoing) in vivo screen, which already revealed several interesting directions for future research in the field of neuronal remodeling. This is, to our knowledge, one of the most extensive direct in vivo CRISPR screens reported to date. Furthermore, we generated a collection of gRNA-expressing fly lines using optimized tools, which provides an invaluable contribution to the fly genetics toolkit, and can be easily utilized for large-scale in vivo screening in various biological contexts.

Due to the technical complexity of mutagenesis-based screens in *Drosophila*, the leading alternative in recent years has been RNAi, and several libraries have been generated (e.g., https://fgr.hms.harvard.edu/fly-in-vivo-rnai, https://stockcenter.vdrc.at/control/library_rnai). While these constitute important resources, genetic knockdown occurs at the mRNA rather than at the DNA level, and is often incomplete[19,35]. Most notably in the fly nervous system, RNAi demonstrates inconsistency that is manifested by variable and unpredictable targeting efficiency within and between different lines (unpublished observations). Since gRNAs

are also known to vary in efficiency[27,36], and due to the 1/3 probability of in-frame indel mutations, tsCRISPR could also potentially suffer from partial or variable penetrance. However, our proof-of-concept experiment demonstrated that phenotypes achieved by tsCRISPR are significantly more penetrant than those induced by RNAi. Moreover, while the vast majority of RNAi lines were ineffective, the same was true for merely a minority the gRNA lines, demonstrating the superiority of tsCRISPR—especially in the context of a screen which aims to minimize false negative results. Still, due to the existence of ineffective gRNAs, we chose to express two different gRNAs per gene, acting as a safety net in case one is inactive.

Importantly, other groups have embarked on similar, ongoing endeavors to construct large-scale gRNA libraries. These include a resource which utilizes either pCFD3 or pCFD4 for not only knockout but also transcriptional activation purposes[37] (https://fgr.hms.harvard.edu/fly-in-vivo-crispr-cas), as well as a pCFD6 collection that allows tissue-specific gRNA expression to reduce ectopic mutagenesis (http://www.crisprflydesign.org/library). While we found pCFD5 (and to a lesser extent pCFD4) to be the most efficient for gene knockout, these valuable complimentary resources can account for genes not included in our library, for overexpression experiments, or when avoiding ectopic targeting is crucial.

Finally, baring in mind that tools for conditional transgene expression are not exclusive to *Drosophila*, tsCRISPR could potentially be expended to other model organisms as well, and therefore holds the promise of profoundly impacting not only fly genetics, but also candidate gene targeting in general.

## Methods

**Ethics.** This study was approved by the Weizmann Institute of Science Recombinant DNA Committee

**Drosophila melanogaster rearing and strains.** All fly strains were reared under standard laboratory conditions at 25 °C on molasses-containing food. Males and females were chosen at random. Unless specifically stated otherwise, the relevant developmental stage is adult, which refers to 3–5 days posteclosion.

*R71G10-GAL4* on the second chromosome was previously generated by our lab[25]. Generation of the *bsk[LL02244]*, *UVRAG[LL03097]* and *plum[Δ1]* alleles was previously described[38,39]. The following lines were obtained from the Bloomington Drosophila Stock Center (BDSC): *nos-Cas9*, *UAS-Cas9.P2* (#54591 and #58986 respectively, both generated by Fillip Port and Simon Bullock); *UAS-Cas9.C* (#54595, generated by Hui-Min Chen and Tzumin Lee); all RNAi lines of the TRiP collection (as listed in Supplementary Table 1); CG9003 RNAi line *TRiP.HMJ23893* (#62439), *R71G10-GAL4* on the third chromosome (#39604) and the *slmb3A1* allele (#65423).

**Selection of genes for the gRNA library.** Genes were selected mainly based on the transcriptional data of developing MB γ neurons[25]. We focused on clusters that displayed a peak of expression at the late larval or early pupal stages—just prior to the onset of pruning at 6 h after puparium formation. Within the pruning-related clusters, genes were sorted based on their expression level, and the higher-expressed genes were given preference. In addition, we chose genes whose expression was prominently affected by perturbing transcription factors that play key roles in MB γ axon pruning, including *EcR*, *Eip75B*, and *Sox14*[25]. In some cases, over-representation of genes of a specific gene family (neurotransmitter receptors, for example) led to inclusion of more genes from that family in the library. In addition, genes that encode key players in major signaling pathways were added (using the KEGG database, https://www.genome.jp/kegg/).

**Design of gRNA sequences.** All gRNA sequences were selected using the Fly-CRISPR algorithm (http://flycrispr.molbio.wisc.edu/), contain 20 nucleotides each (PAM excluded), and are predicted to have zero off-targets.

For each of the nine genes in the proof-of-concept screen, three different gRNA sequences were selected, all within the coding region of the gene, as adjacent as possible to the translation initiation site (Supplementary Table 1).

For each gene in the large-scale gRNA-library, two different gRNA sequences were selected, both within the coding region and as adjacent as possible to the translation initiation site—but not overlapping each other (Supplementary Data 1). In the case of multiple isoforms, the coding region common to all isoforms was used. In rare cases (when impossible otherwise), a selected sequence targets only

part of the gene's isoforms. Sequences with high GC content in the PAM-adjacent region were highly preferred (specifically, at least four GCs out of the six nucleotides in the 3′ end of the 20-nucleotide sequence).

**Generation of transgenic constructs and transgenic flies.** For the proof-of-concept screen, each individual gRNA sequence was cloned into the pCFD3 plasmid (Addgene #49410). pCFD3 was digested with BbsI and then ligated with annealed oligonucleotides containing the 20-nucleotide gRNA sequence[14] (see Supplementary Table 2).

For comparison of different gRNA-expression plasmids, four gRNA sequences were selected (two for each gene) as follows:

Mamo:

1. 5′-AGTACGAGGAACAAGCCGAG 2. 5′-GCAGTGAGCACTACTGCTTG
Plum:

1. 5′-CAATCAATTGAATCACAAAG 2. 5′-GTTCTTCGGTTGGGCGACGG
Cloning of both gRNAs (per gene) into pCFD4[14] (Addgene #49411) was done using the transfer PCR (TPCR) method[40], and into pCFD5[15] (Addgene #73914) or pCFD6[15] (Addgene #73915) using Restriction-Free (RF) cloning[41]. Only the first gRNA sequence of the two was cloned into pCFD3, using restriction and ligation (see Supplementary Table 2)

For the large-scale gRNA library, two gRNA sequences per gene were cloned into either the pCFD4 or pCFD5 plasmids. Cloning into pCFD4 was done using TPCR[40], enabling high-throughput generation of plasmids. Due to repetitive sequences, TPCR was inefficient in cloning gRNAs into pCFD5, and therefore we used the services of BioBasic (https://www.biobasic.com/) via Syntezza Bioscience (https://syntezza.com/).

gRNA-harboring constructs were injected to *Drosophila* embryos and integrated into attP landing sites using the φC31 system, as follows:

All 27 constructs of the proof-of-concept screen, as well as all Mamo-gRNA and Plum-gRNA constructs (pCFD3/4/5/6) were integrated into [*M3xP3-RFP.attP]ZH-86Fb* (86Fb) on the third chromosome.

Constructs of the large-scale gRNA fly collection were integrated into either 86Fb on the third chromosome, or *P[y[ + t7.7]CaryP]attP40* (attP40) on the second chromosome—as listed in Supplementary Data 1.

Injections were performed in-house, or as services by either Bestgene (https://www.thebestgene.com/) or Rainbow Transgenic Flies (https://www.rainbowgene.com/).

Our entire collection of gRNA-harboring plasmids and transgenic flies is available to the community via the public repositories of the Drosophila Genomics Resource Center (DGRC, https://dgrc.bio.indiana.edu/Home) and the Bloomington Drosophila Stock Center (BDSC, https://bdsc.indiana.edu/), respectively.

**Generation of the CG9003[indel] mutant.** Transgenic flies expressing CG9003-gRNAx2[pCFD4] were crossed to flies expressing *nos-Cas9*. Flies containing both the gRNAs and *nos-Cas9* were crossed to a balancer line, and single male offspring were then crossed to a balancer line and checked for the presence of an indel using specific primers (see Supplementary Table 2).

The resulting indel is a deletion of 4 nucleotides and insertion of 74 others, 15 nucleotides downstream of the translation initiation site of the coding sequence of isoform C. The first 3 nucleotides of the 74 encode a stop codon (TAA), resulting in predicted truncation of the protein upstream of all putative domains in all isoforms (Supplementary Fig. 4b).

**Generation of MARCM clones.** MARCM[26] clones were generated by a 1 h heat-shock (37 °C) of newly hatched larvae, 24 h after egg laying. Brains were dissected at the adult stage.

**Immunostaining and imaging.** *Drosophila* brains were dissected in cold ringer solution, fixed using 4% paraformaldehyde for 20 min at room temperature (RT), and then washed in phosphate buffer with 0.3% Triton-X (PBT; 3× immediate washes followed by 3 × 20-min washes). Non-specific staining was blocked using 5% heat inactivated goat serum in PBT, and brains were then subjected to primary antibody staining overnight at 4 °C. Primary antibodies included chicken anti-GFP 1:500 (GFP-1020; AVES), mouse anti-FasII 1:25 (1D4; DSHB), mouse anti-EcRB1 1:25 (AD4.4; DSHB), rabbit anti-active-JNK (pJNK) 1:200 (V7931; Promega) and rabbit anti-Mamo 1:5000[25]. Brains were rinsed (x3) then washed with PBT (3 × 20-min), stained with secondary antibodies for 2 h at RT, and washed again. Secondary antibodies included FITC donkey anti-chicken 1:300 (703-095-155; Jackson immunoresearch), Alexa fluor 647 goat anti-mouse 1:300 (A-21236; Invitrogen) and Alexa fluor 647 goat anti-rabbit 1:300 (A-21236; Invitrogen). When staining for Mamo, DAPI 1:1000 (D1306; Invitrogen) was added for 15 min and then rinsed three times prior to mounting. Brains were mounted on Slowfade (S-36936; Invitrogen) and imaged on Zeiss LSM 800 confocal microscope. Images were processed with ImageJ (NIH).

**Quantification and statistical analysis.** In Fig. 1d–e, phenotypic penetrance was defined per each gRNA or RNAi line as the percentage of hemispheres that displayed a detectable MB pruning defect (see also Supplementary Table 1). Groups were compared using two-tailed Mann–Whitney $U$ test.

Quantification of pruning defect severity of MB γ neurons was performed by phenotypic ranking[38,42]. Images were blindly ranked by two independent investigators according to increasing phenotypic severity, determined based on the amount of dorsally projecting γ axons (i.e., GFP-labeled axons that do not coincide with the FasII-stained axonal bundle). Results from both rankers were compared to each other using paired Wilcoxon signed-rank test and did not differ significantly (see p values and plots in Supplementary Fig. 2b and Supplementary Fig. 4e). The results from each ranker individually were analyzed using either Kruskal-Wallis test followed by pairwise Wilcoxon test with FDR correction (Fig. 2c, Supplementary Fig. 2a), or by two-tailed Mann–Whitney U test (Fig. 4d, Supplementary Fig. 4d), and p values are reported within the relevant figure legends. The combined results from both rankers were analyzed using a linear mixed effects model (accounting for the ranker as a random effect), and p values are reported within the results section.

Quantification of Mamo-immunoreactivity (Fig. 2d, Supplementary Fig. 3d) was performed using a custom-built FIJI macro. In brief, GFP was used to define the region of interest, in which DAPI staining was used to segment the γ cell bodies. For each cell body, Mamo staining mean intensity was measured. In each hemisphere, measurements were taken in five separate slices (approximately 3 μm apart). Finally, we determined the proportion of cells that lost immunoreactivity for each hemisphere (calculated as less than 2-fold of the background staining, determined independently for each image). Groups were compared either by one-way Anova followed by Tukey's test for multiple comparisons (Fig. 2d), or by two-tailed Student's t test (Supplementary Fig. 3d).

Specific p values are indicated in the relevant figure legends.

**Detailed *Drosophila* genotypes.** Genotype abbreviations: R71G10 is GMR71G10-GAL4, 201Y is 201Y-GAL4, Repo is Repo-GAL4, CD8 is 10XUAS-mCD8::GFP, hsFlp is y,w,hsFlp22, Cas9.C is UAS-Cas9.C, Cas9.P2 is UAS-Cas9.P2. 19A, 40A, and 82B are FRTs on the X, second and third chromosomes, respectively. For gRNA-lines, the name of the gene and gRNA-expression plasmid are listed—the promoter varies depending on the plasmid (U6:3 for both pCFD3 and pCFD5, U6:1 + U6:3 for pCFD4, UAS for pCFD6), and gRNAx2 means that two different gRNAs were expressed per gene. The reported genotype is female, but males and females were used interchangeably.

Figure 1c. (R71G10 > Cas9.C):
    (WT) y,w;R71G10,CD8/+; Cas9.C/+
    (Plum) y,w/y,v;R71G10,CD8/+; Cas9.C/Plum-gRNA$^{\text{pCFD3\#1}}$
    (UVRAG) y,w/y,v;R71G10,CD8/+; Cas9.C/UVRAG-gRNA$^{\text{pCFD3\#3}}$
    (Bsk) y,w/y,v;R71G10,CD8/+; Cas9.C/Bsk-gRNA$^{\text{pCFD3\#3}}$
Figure 1c. (MARCM clones):
    (WT) hsFlp,CD8/+; 201Y,CD8/+; 82B/82B,GAL80
    (Plum) hsFlp,CD8/+; 201Y,CD8/+; 82B,plum$^{\Delta1}$/82B,GAL80
    (UVRAG) hsFlp,CD8/+; UVRAG$^{\text{LL3097}}$,40A/GAL80,40A,CD8,201Y(Bsk) hsFlp,CD8/+ ; bsk$^{\text{LL02244}}$,40A/GAL80,40A,CD8,201Y
Figure 1c. (R71G10 > RNAi):
    (WT) y,w;R71G10/+; CD8/+
    (Plum) y,w/y,v;R71G10/TRiP.HMC05055;CD8/+ (UVRAG) y,w/y,v;R71G10/+; CD8/TRiP.HMS01357
    (Bsk) y,w/y,v;R71G10/TRiP.HMS04479;CD8/+
Figure 2a. (pCFD3) y,w/y,v;R71G10,CD8/+; Cas9.C/Plum-gRNA$^{\text{pCFD3}}$
    (pCFD4) y,w/y,v;R71G10,CD8/+; Cas9.C/Plum-gRNAx2$^{\text{pCFD4}}$
    (pCFD5) y,w/y,v;R71G10,CD8/+; Cas9.C/Plum-gRNAx2$^{\text{pCFD5}}$
    (pCFD6) y,w;R71G10,CD8/+; Cas9.C/Plum-gRNAx2$^{\text{pCFD6}}$
Figure 2b. (pCFD3) y,w/y,v;R71G10,CD8/+; Cas9.C/Mamo-gRNA$^{\text{pCFD3}}$
    (pCFD4) y,w/y,v;R71G10,CD8/+; Cas9.C/Mamo-gRNAx2$^{\text{pCFD4}}$
    (pCFD5) y,w/y,v;R71G10,CD8/+; Cas9.C/Mamo-gRNAx2$^{\text{pCFD5}}$
    (pCFD6) y,w;R71G10,CD8/+; Cas9.C/Mamo-gRNAx2$^{\text{pCFD6}}$
Figure 3b. (WT) y,w;R71G10,CD8/+; Cas9.C/+(CG9003) y,w/y,v;R71G10, CD8/+;Cas9.C/CG9003-gRNAx2$^{\text{pCFD4}}$
    (Pli) y,w/y,v;R71G10,CD8/+; Cas9.C/Pli-gRNAx2$^{\text{pCFD4}}$
    (CG34354) y,w/y,v;R71G10,CD8/+; Cas9.C/CG34353-gRNAx2$^{\text{pCFD4}}$
    (Hppy) y,w/y,v;R71G10,CD8/+; Cas9.C/Hppy-gRNAx2$^{\text{pCFD4}}$
    (Mmp2) y,w/y,v;R71G10,CD8/+; Cas9.C/Mmp2-gRNAx2$^{\text{pCFD4}}$
    (Dop1R2) y,w/y,v;R71G10,CD8/+; Cas9.C/Dop1R2-gRNAx2$^{\text{pCFD4}}$
    (danr) y,w/y,v;R71G10,CD8/+; Cas9.C/danr-gRNAx2$^{\text{pCFD4}}$
Figure 4a. (WT) y,w;+/+; R71G10,CD8/+(CG9003$^{\text{indel/indel}}$) y,w;CG9003$^{\text{indel}}$/ CG9003$^{\text{indel}}$;R71G10,CD8/+
Figure 4b. (WT) y,w,hsFLP,CD8/+; 201Y,CD8/+; 82B/82B,GAL80
    (slmb3A1) hsFLP,CD8/+; 201Y,CD8/+; 82B,slmb3A1/82B,GAL80
Figure 4c. (slmb$^{\text{3A1/+}}$) y,w;+/+; R71G10,CD8/slmb3A1
    (slmb$^{\text{3A1/+}}$ + CG9003$^{\text{indel/indel}}$) y,w;CG9003$^{\text{indel}}$/CG9003$^{\text{indel}}$;R71G10, CD8/slmb3A1
Supplementary Fig. 1a. (R71G10 > Cas9.C):
    (WT) y,w;R71G10,CD8/+; Cas9.C/+
    (USP) y,w/y,v;R71G10,CD8/+; Cas9.C/USP-gRNA$^{\text{pCFD3\#1}}$
    (Rpn6) y,w/y,v;R71G10,CD8/+; Cas9.C/Rpn6-gRNA$^{\text{pCFD3\#3}}$
    (Uba1) y,w/y,v;R71G10,CD8/+; Cas9.C/Uba1-gRNA$^{\text{pCFD3\#3}}$
Supplementary Fig. 1a (R71G10 > RNAi):
    (WT) y,w;R71G10/+; CD8/+

    (USP) y,w/y,v;R71G10/+; CD8/TRiP.HMS01620
    (Rpn6) y,w/y,v;R71G10/+; CD8/TRiP.JF03317
    (Uba1) y,w/y,v;R71G10/+; CD8/TRiP.GL00491
Supplementary Fig. 1b. (WT) y,w;R71G10,CD8/+; Cas9.C/+
    (Bsk) y,w/y,v;R71G10,CD8/+; Cas9.C/Bsk-gRNA$^{\text{pCFD3\#1}}$
Supplementary Fig. 1c. (WT) y,w;R71G10,CD8/+; Cas9.P2/+
    (EcR) y,w/y,v;R71G10,CD8/EcR-gRNA$^{\text{pCFD3\#3}}$;Cas9.P2/+
Supplementary Fig. 1d. (WT) y,w;CD8/+; Repo/Cas9.P2
    (EcR) y,w/y,v;CD8/EcR-gRNA$^{\text{pCFD3\#3}}$;Repo/Cas9.P2
Supplementary Fig. 3a. (WT) y,w;R71G10,CD8/+; Cas9.P2/+
    (EcR) y,w/y,v;R71G10,CD8/EcR-gRNA$^{\text{pCFD3\#3}}$;Cas9.P2/+(Mov34) y,w/y,v; R71G10,CD8/+; Cas9.P2/Mov34-gRNA$^{\text{pCFD3\#1}}$
Supplementary Fig. 3b. (R71G10 > Cas9.C) y,w/y,v;R71G10,CD8/+;Cas9.C/ Bsk-gRNA$^{\text{pCFD3\#3}}$
    (R71G10 > Cas9.P2) y,w/y,v;R71G10,CD8/ +;Cas9.P2/Bsk-gRNA$^{\text{pCFD3\#3}}$
Supplementary Fig. 3c. (R71G10 > Cas9.C) y,w/y,v;R71G10,CD8/+; Cas9.C/ Mamo-gRNAx2$^{\text{pCFD4}}$
    (R71G10 > Cas9.P2) y,w/y,v;R71G10,CD8/+; Cas9.P2/Mamo-gRNAx2$^{\text{pCFD4}}$
Supplementary Fig. 4a. (WT) y,w;R71G10/+; CD8/+
    (CG9003 RNAi) y,w/y,v;R71G10/ TRiP.HMJ23893;CD8/+
Supplementary Fig. 4c. (WT) y,w;+ / + ;R71G10,CD8/ + (CG9003$^{\text{indel/indel}}$) y,w; CG9003$^{\text{indel}}$/CG9003$^{\text{indel}}$;R71G10,CD8/ +

**Reporting summary.** Further information on research design is available in the Nature Research Reporting Summary linked to this article.

## Data availability
Previously constructed plasmids Addgene #49410, Addgene #49411, Addgene #73914, and Addgene #73915 were obtained from Addgene. Newly constructed plasmids are available via the *Drosophila* Genomics Resource Center (https://dgrc.bio.indiana.edu/ Home). Transgenic flies are available via the Bloomington *Drosophila* Stock Center (https://bdsc.indiana.edu). The data sets generated and analyzed during the current study are available from the corresponding author on reasonable request. The source data underlying Figs. 1d, e, 2c, d, 3c and 4d and Supplementary Figs. 2a, 3d, and 4d are provided as a Source Data file.

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

# ARTICLE

15. Port, F. & Bullock, S. L. Augmenting CRISPR applications in Drosophila with tRNA-flanked sgRNAs. *Nat. Methods* **13**, 852–854 (2016).

16. Xue, Z. et al. CRISPR/Cas9 mediates efficient conditional mutagenesis in Drosophila. *G3* **4**, 2167–2173 (2014).

17. Huynh, N., Zeng, J., Liu, W. & King-Jones, K. A drosophila CRISPR/Cas9 toolkit for conditionally manipulating gene expression in the prothoracic gland as a test case for polytene tissues. *G3* **8**, 3593–3605 (2018).

18. Poe A. R., et al. Robust CRISPR/Cas9-mediated tissue specific mutagenesis reveals gene redundancy and perdurance in drosophila. *Genetics* **211**, 459–472 (2018).

19. Dietzl, G. et al. A genome-wide transgenic RNAi library for conditional gene inactivation in Drosophila. *Nature* **448**, 151–156 (2007).

20. Schuldiner, O. & Yaron, A. Mechanisms of developmental neurite pruning. *Cell Mol. Life Sci.* **72**, 101–119 (2015).

21. Sekar, A. et al. Schizophrenia risk from complex variation of complement component 4. *Nature* **530**, 177–183 (2016).

22. Hong, S. et al. Complement and microglia mediate early synapse loss in Alzheimer mouse models. *Science* **352**, 712–716 (2016).

23. Yu, F. & Schuldiner, O. Axon and dendrite pruning in Drosophila. *Curr. Opin. Neurobiol.* **27**, 192–198 (2014).

24. Yaniv, S. P. & Schuldiner, O. A fly's view of neuronal remodeling. *Wiley Inter. Rev. Dev. Biol.* **5**, 618–635 (2016).

25. Alyagor, I. et al. Combining developmental and perturbation-seq uncovers transcriptional modules orchestrating neuronal remodeling. *Dev. Cell* **47**, 38–52 (2018). e36.

26. Lee, T. & Luo, L. Mosaic analysis with a repressible cell marker for studies of gene function in neuronal morphogenesis. *Neuron* **22**, 451–461 (1999).

27. Ren, X. et al. Enhanced specificity and efficiency of the CRISPR/Cas9 system with optimized sgRNA parameters in Drosophila. *Cell Rep.* **9**, 1151–1162 (2014).

28. Lee, T., Marticke, S., Sung, C., Robinow, S. & Luo, L. Cell-autonomous requirement of the USP/EcR-B ecdysone receptor for mushroom body neuronal remodeling in Drosophila. *Neuron* **28**, 807–818 (2000).

29. Bornstein, B. et al. Developmental axon pruning requires destabilization of cell adhesion by JNK signaling. *Neuron* **88**, 926–940 (2015).

30. Watts, R. J., Hoopfer, E. D. & Luo, L. Axon pruning during *Drosophila metamorphosis*: evidence for local degeneration and requirement of the ubiquitin-proteasome system. *Neuron* **38**, 871–885 (2003).

31. Wong J. J. L., et al. A cullin1-based SCF E3 ubiquitin ligase targets the InR/PI3K/TOR pathway to regulate neuronal pruning. *PloS Biol.* **11**, e1001657 (2013).

32. Cardozo, T. & Pagano, M. The SCF ubiquitin ligase: insights into a molecular machine. *Nat. Rev. Mol. Cell Biol.* **5**, 739–751 (2004).

33. Buster, D. W. et al. SCFSlimb ubiquitin ligase suppresses condensin II-mediated nuclear reorganization by degrading Cap-H2. *J. Cell Biol.* **201**, 49–63 (2013).

34. Yaniv, S. P., Issman-Zecharya, N., Oren-Suissa, M., Podbilewicz, B. & Schuldiner, O. Axon regrowth during development and regeneration following injury share molecular mechanisms. *Curr. Biol.* **22**, 1774–1782 (2012).

35. Booker, M. et al. False negative rates in Drosophila cell-based RNAi screens: a case study. *BMC Genom.* **12**, 50 (2011).

36. Moreno-Mateos, M. A. et al. CRISPRscan: designing highly efficient sgRNAs for CRISPR-Cas9 targeting in vivo. *Nat. Methods* **12**, 982–988 (2015).

37. Lin, S., Ewen-Campen, B., Ni, X., Housden, B. E. & Perrimon, N. In vivo transcriptional activation using CRISPR/Cas9 in drosophila. *Genetics* **201**, 433–442 (2015).

38. Schuldiner, O. et al. piggyBac-based mosaic screen identifies a postmitotic function for cohesin in regulating developmental axon pruning. *Dev. Cell* **14**, 227–238 (2008).

39. Yu, X. M. et al. Plum, an immunoglobulin superfamily protein, regulates axon pruning by facilitating TGF-beta signaling. *Neuron* **78**, 456–468 (2013).

40. Erijman, A., Shifman, J. M. & Peleg, Y. A single-tube assembly of DNA using the transfer-PCR (TPCR) platform. *Methods Mol. Biol.* **1116**, 89–101 (2014).

41. Unger, T., Jacobovitch, Y., Dantes, A., Bernheim, R. & Peleg, Y. Applications of the restriction free (RF) cloning procedure for molecular manipulations and protein expression. *J. Struct. Biol.* **172**, 34–44 (2010).

42. Issman-Zecharya, N. & Schuldiner, O. The PI3K class III complex promotes axon pruning by downregulating a Ptc-derived signal via endosome-lysosomal degradation. *Dev. Cell* **31**, 461–473 (2014).

## Acknowledgements

We thank E. Arama for joint fundraising and discussions; R. Rotkopf for help with the statistical analysis; L. Luo for assistance and discussions; the Bloomington Stock Center and the Developmental Studies Hybridoma Bank for reagents; and members of the Schuldiner lab—especially S. Yaniv, N. Marmor-Kollet and N. Zecharya—for technical assistance, discussions and critical reading of the manuscript. This work was supported by the Estate of Olga Klein-Astrachan, the European Research Council (erc CoG "AxonGrowth"), the David and Fela Shapell Family Center for Genetic Disorders Research, and the Adelis Foundation. O.S. is the incumbent of the Professor Erwin Neter Professorial Chair of Cell Biology.

## Author contributions

H.M designed, performed and analyzed the experiments and wrote the manuscript; E.M and V.B provided the technical assistance and performed specific experiments; O.M. developed image quantification methods; I.A. provided the important data and helped with its analysis; N.S.G. and T.U. provided the high-throughput cloning services; D.L. provided the technical assistance with embryo injection; O.S. led the project, designed the experiments, interpreted the results and wrote the paper.

## Additional information

**Competing interests:** The authors declare no competing interests.

