## [Peer Review File · Nature Communications]

Reviewers' Comments:

Reviewer #1:

Remarks to the Author:

Although the CRISPR technology has been widely used in *Drosophila* for creating germline mutations and for genome editing, large scale in vivo genetic screens based on CRISPR has not been reported in *Drosophila*. In this manuscript, the authors reported the comparison of some of the previously reported approaches for somatic CRISPR mutagenesis, which they termed tsCRISPR, in developmental pruning of mushroom body neurons, and described their efforts in developing gRNA plasmid and transgene libraries. The comparison of various gRNA construction methods will likely be useful for other similar genetic screens and the gRNA libraries will likely be valuable resources for the *Drosophila* community.

The work described in the manuscript was nicely done. However, I found a few issues in the writing, experimental design, and result interpretation. I would be supportive for its acceptance by Nature Communications if these issues are resolved.

Major points:

1. As a manuscript focusing on resource development, it should provide sufficient technical information in the main text so that the readers can understand the approaches. I found the information about various CRISPR approaches in several places is insufficient. The first is when tsCRISPR is first introduced (lines 46-51). It should have a brief summary of the various approaches reported, including how Cas9 and gRNAs are delivered in each approach. Second, when gRNA construction was described, the authors mentioned plasmid names (pCFD3-pCFD5) without clarifying the major features and differences of these plasmids (lines 89, 119, 127). This makes it difficult to understand the source of the differences in their efficiencies. Lastly, in the last paragraph, three other resources of gRNA libraries were mentioned. Since the strategy for constructing gRNA libraries is important for the CRISPR efficacy, it would be helpful to include a brief summary on the differences among the approaches used. It would be also helpful to add a discussion on potential pros and cons of these approaches and/or potential ways to improve the libraries.
2. The authors found that 6 gRNA lines resulted in lethal crosses in their pilot study and concluded that the lethality is due to Cas9 toxicity. However, in my opinion, this conclusion is wrong. Cas9 toxicity should be independent of the gRNA target site and therefore should be present in all crosses. The more likely explanation is that the Cas9 has leaky expression in other tissues. It was the mutations in those other tissues that caused the animal lethality. Similarly, the conclusion about the toxicity of Cas9 in the comparison of UAS-Cas9.P2 and UAS-Cas9.C is likely wrong, too. Most likely UAS-Cas9.P2 is less leaky and expressed at a lower level than UAS-Cas9.C.
3. Related to the last point, since the authors supported the use of UAS-Cas9.C in large scale genetic screens, it is important to understand the nature of the leakiness of UAS-Cas9.C. At the minimum, the authors should determine if the leakiness is Gal4 dependent or not.
4. Since the authors developed the plasmid and transgenic fly libraries with the broader *Drosophila* community in mind, there should be information on how these reagents will be distributed, for example, whether the stocks will be distributed to a stock center.

Minor points:

1. Line 92, "The observed pruning defects were highly similar to those of germline mutant clones of the same genes" is confusing. How would germline mutant clones show neuronal pruning phenotypes? I guess the author tried to say that those are neuronal clones carrying mutations derived from the germline. The wording needs to be changed.
2. Line 59, "inconsistent efficiency" is confusing. Does it imply low efficiency (and thus weaker

phenotype), or variability among samples, or variability among different RNAi lines? tsCRISPR theoretically could have these issues as well.

3. There are no details or reference about how phenotypic severity is ranked.

4. Line 324 and 325, description of Supplementary Fig. 3d does not match the figure.

5. Line 134, the Mov34 phenotype is not obvious in the figure.

6. Some references are missing in the text: high-throughput CRISPR screens (line 37); EcR and Bsk (line 105); MoV34 (line 131); Skp1-Cullin-F-BOX (SCF) E3 ligase complex (line 171); Slmb (line 190).

Reviewer #2:

Remarks to the Author:

In the manuscript by Meltzer and colleagues, the authors use the mushroom body gamma neuron system in *Drosophila* to experimentally test various guide-RNA conditions when using CRISPR/Cas9 for a tissue specific mutant screen. The authors present solid evidence that tissue specific CRISPR can be used as an effective method for screening, likely more reliable and penetrant than using RNAi. The data also strongly supports the use of pCFD5 (and pCFD4 to a lesser degree) for consistently eliminating gene function in the central brain. The authors generated a library of 900 gRNA plasmids and 200 gRNA transgenic lines, and successfully performed a large mutant screen to identify novel genes involved in gamma-neuron pruning. They also generated germ-line mutations using the same gRNA transgenic lines. This work represents a clear way forward for conducting similar tissue-specific screens. It also represents how most genetic screen will be conducted in the future: using gRNA lines to perform direct forward genetics screens (likely based on temporal and tissue specific expression data). The 200 gRNA transgenic library (and growing) will be a highly utilized resource. I do not have any major concerns with the work.

Minor comments:

1. The plasmids and gRNA transgenic lines are valuable resources. It was not clear if these would be available from the group's website, or deposited en masse to repositories (ie, Addgene, Bloomington, etc). What are the plans for distribution?

2. Similarly, other groups will add to this effort (generate plasmids and/or gRNA transgenics), and it would be best to coordinate efforts. Do the authors have a plan for this? If so, it might be useful to briefly mention it in the main text or in the methods.

3. Line 143. "next step was to apply it in large-scale and screen" might be better written as "next step was to apply it in a large scale screen"

4. Figure 1h. "Phenotypic severity". How quantitative vs qualitative is this measurement? For example, if 3 people scored these phenotypes, how closely would those scores agree? Some discussion of this in the Methods would be helpful to gauge the accuracy of these evaluations.

5. Figure Legends. Please include the statistical test used for each calculated p-value. These are in the Methods, but should also be included in the legends.

6. Figure 2c legend. Please include the reference for these heatmaps in the legend. I think it is Ref. 19 Alyagor et al? Or if they were generated for this work, make that clearer in the main text.

Point-by-point response to reviewers:

General changes in the manuscript:

We thank the reviewers for their positive assessment of our work and appreciate their constructive comments and suggestions. The extremely abbreviated format of the original manuscript resulted from constraints of our first submission to another journal from the Nature family. Indeed, in the revised manuscript we now significantly changed the format to include more information, including the important points that the reviewers requested, and additionally split the figures into four main figures plus four supplementary figures.

The dissemination of the tools and resources that we have generated are indeed a central part of our work and we are happy to report that the Bloomington Drosophila Stock Center has agreed to host and distribute our fly stocks (that has by now increased to ~300 lines) while the Drosophila Genomics Resource Center has agreed to do the same for our plasmid collection.

We trust that the editor and reviewers will appreciate the changes we have done to the manuscript and will support publication in Nature Communications.

A point-by-point reply to the reviewers is attached below – the original comments are in blue.

Reviewer #1 (Remarks to the Author):

Although the CRISPR technology has been widely used in Drosophila for creating germline mutations and for genome editing, large scale in vivo genetic screens based on CRISPR has not been reported in Drosophila. In this manuscript, the authors reported the comparison of some of the previously reported approaches for somatic CRISPR mutagenesis, which they termed tsCRISPR, in developmental pruning of mushroom body neurons, and described their efforts in developing gRNA plasmid and transgene libraries. The comparison of various gRNA construction methods will likely be useful for other similar genetic screens and the gRNA libraries will likely be valuable resources for the Drosophila community.

The work described in the manuscript was nicely done. However, I found a few issues in the writing, experimental design, and result interpretation. I would be supportive for its acceptance by Nature Communications if these issues are resolved.

Major points:

1. As a manuscript focusing on resource development, it should provide sufficient technical information in the main text so that the readers can understand the approaches. I found the information about various CRISPR approaches in several places is insufficient. The first is when tsCRISPR is first introduced (lines 46-51). It should have a brief summary of the various approaches reported, including how Cas9 and gRNAs are delivered in each approach. Second, when gRNA construction was described, the authors mentioned plasmid names (pCFD3-pCFD5) without clarifying the major features and differences of these plasmids (lines 89, 119, 127). This makes it difficult to understand the source of the differences in their efficiencies.

Thanks for these important points. We have now amended the text to better describe current existing approaches. In lines (new line number) 50-63 we now include information about germ-line expression of Cas9 as well as tissue specific expression using either the binary Gal4-UAS system or by direct enhancer-Cas9 fusion.

Likewise, we now elaborate more on the differences between the different pCFD plasmids – see lines 103-105 regarding pCFD3; 137-141 regarding pCFD4-5; 153-154 regarding pCFD6.

Lastly, in the last paragraph, three other resources of gRNA libraries were mentioned. Since the strategy for constructing gRNA libraries is important for the CRISPR efficacy, it would be helpful to include a brief summary on the differences among the approaches used. It would be also helpful to add a discussion on potential pros and cons of these approaches and/or potential ways to improve the libraries.

We rewrote this section. The problem with providing a full scale comparison is that these studies are mostly unpublished and therefore some of the specifics still unknown. In the case of the Harvard collection (Perrimon), they have moved from pCFD3 to pCFD4, which contains two gRNAs in contrast to one, presumably for reasons similar to those found in our study. Their collection also includes the CRISPRa approach which is unique as it provides means to activate rather than knock out gene function. The DKFZ (Port, Boutros, Bullock) collection is based on the pCFD6. We now describe these differences in greater details and suggest how they could become complementary (lines 283-292).

2. The authors found that 6 gRNA lines resulted in lethal crosses in their pilot study and concluded that the lethality is due to Cas9 toxicity. However, in my opinion, this conclusion is wrong. Cas9 toxicity should be independent of the gRNA target site and therefore should be present in all crosses. The more likely explanation is that the Cas9 has leaky expression in other tissues. It was the mutations in those other tissues that caused the animal lethality. Similarly, the conclusion about the toxicity of Cas9 in the comparison of UAS-Cas9.P2 and UAS-Cas9.C is likely wrong, too. Most likely UAS-Cas9.P2 is less leaky and expressed at a lower level than UAS-

Cas9.C.

We completely agree with this reviewer and apologize for our confusion. Indeed, we found that Cas9.C lethality is due to its leaky expression regardless of Gal4 expression while Cas9.P2 was lethal only when combined with Gal4 driver. In both cases, we agree that lethality is likely due to non-MB specific deletions of Mov34 and EcR. We now discuss this in more accuracy in the text – in lines 162-192.

3. Related to the last point, since the authors supported the use of UAS-Cas9.C in large scale genetic screens, it is important to understand the nature of the leakiness of UAS-Cas9.C. At the minimum, the authors should determine if the leakiness is Gal4 dependent or not.

See reply to previous comment. In short, UAS-Cas9.C induced lethality does not depend on Gal4 (as previously shown in Port and Bullock, 2016, now properly referenced) in the case of Mov34 and EcR. However, it is important to state that in our large scale screen, Cas9.C lethality was observed in only around 1% of the cases (as discussed in the text).

4. Since the authors developed the plasmid and transgenic fly libraries with the broader Drosophila community in mind, there should be information on how these reagents will be distributed, for example, whether the stocks will be distributed to a stock center.

We are happy to report that the Bloomington Drosophila Stock Center has agreed to host and distribute our fly stocks (now increased to ~300 lines) while the Drosophila Genomics Resource Center has agreed to do the same for our plasmid collection.

Minor points:

1. Line 92, “The observed pruning defects were highly similar to those of germline mutant clones of the same genes” is confusing. How would germline mutant clones show neuronal pruning phenotypes? I guess the author tried to say that those are neuronal clones carrying mutations derived from the germline. The wording needs to be changed.

We changed the wording according to the reviewers' suggestions.

2. Line 59, “inconsistent efficiency” is confusing. Does it imply low efficiency (and thus weaker phenotype), or variability among samples, or variability among different RNAi lines? tsCRISPR theoretically could have these issues as well.

We now expanded this section and moved it to the discussion section lines 264-282. Obviously tsCRISPR could suffer from the same issues but our pilot screen, where we compared tsCRISPR to RNAi suggests it is much more robust in gene targeting efficiency.

3. There are no details or reference about how phenotypic severity is ranked.

We now include more information about the ranking including a reference of Issman and Schuldiner 2014. Importantly, however, the ranking was now repeated by an independent investigator – the results were extremely similar – both ranking experiments are shown and discussed. The actual images and ranking are provided in the source data.

4. Line 324 and 325, description of Supplementary Fig. 3d does not match the figure.

Thanks for spotting this error, which was fixed.

5. Line 134, the Mov34 phenotype is not obvious in the figure.

Indeed the phenotype is weak, and we added a note describing it as such. Additionally, we now added a new panel in Fig. S3a to show the CD8 channel alone.

6. Some references are missing in the text: high-throughput CRISPR screens (line 37); EcR and Bsk (line 105); MoV34 (line 131); Skp1-Cullin-F-BOX (SCF) E3 ligase complex (line 171); Slmb (line 190).

The limited number of references was a limitation of the original format of our submission. This was now fixed.

Reviewer #2 (Remarks to the Author):

In the manuscript by Meltzer and colleagues, the authors use the mushroom body gamma neuron system in Drosophila to experimentally test various guide-RNA conditions when using CRISPR/Cas9 for a tissue specific mutant screen. The authors present solid evidence that tissue specific CRISPR can be used as an effective method for screening, likely more reliable and penetrant than using RNAi. The data also strongly supports the use of pCFD5 (and pCFD4 to a lesser degree) for consistently eliminating gene function in the central brain. The authors generated a library of 900 gRNA plasmids and 200 gRNA transgenic lines, and successfully performed a

large mutant screen to identify novel genes involved in gamma-neuron pruning. They also generated germ-line mutations using the same gRNA transgenic lines. This work represents a clear way forward for conducting similar tissue-specific screens. It also represents how most genetic screen will be conducted in the future: using gRNA lines to perform direct forward genetics screens (likely based on temporal and tissue specific expression data). The 200 gRNA transgenic library (and growing) will be a highly utilized resource. I do not have any major concerns with the work.

Minor comments:

1. The plasmids and gRNA transgenic lines are valuable resources. It was not clear if these would be available from the group's website, or deposited en masse to repositories (ie, Addgene, Bloomington, etc). What are the plans for distribution?

We are happy to report that the Bloomington Drosophila Stock Center has agreed to host and distribute our fly stocks (now increased to ~300 lines) while the Drosophila Genomics Resource Center has agreed to do the same for our plasmid collection.

2. Similarly, other groups will add to this effort (generate plasmids and/or gRNA transgenics), and it would be best to coordinate efforts. Do the authors have a plan for this? If so, it might be useful to briefly mention it in the main text or in the methods.

This is an interesting idea. We don't currently have a plan for coordinating this effort. Despite the fact that this manuscript focuses on a method and resource, its main purpose was to expand the tools available to our lab which is focused on mechanistic studies of neuronal remodeling. Therefore, we anticipate that coordination between the two big, and well-funded, centers – Harvard and DKFZ, and potentially also smaller labs like ourselves, is important and will likely be based on the accumulation of data and information from multiple studies.

3. Line 143. "next step was to apply it in large-scale and screen" might be better written as "next step was to apply it in a large scale screen"

Thanks, corrected.

4. Figure 1h. "Phenotypic severity". How quantitative vs qualitative is this measurement? For example, if 3 people scored these phenotypes, how closely would those scores agree? Some discussion of this in the Methods would be helpful to gauge the accuracy of these evaluations.

We now expanded this section - rankings were performed by two independent investigators with no significant differences between them. Comparison between these ranking is provided in Figure S2b + S4e, individual p- and W-values as well as the combined values are provided in the text and legend and explained in the methods section. Finally, the actual images and ranks are now provided in the source file.

5. Figure Legends. Please include the statistical test used for each calculated p-value. These are in the Methods, but should also be included in the legends.

Done, thanks!

6. Figure 2c legend. Please include the reference for these heatmaps in the legend. I think it is Ref. 19 Alyagor et al? Or if they were generated for this work, make that clearer in the main text.

The reviewer is correct – we now added Alyagor et al as the correct reference for the heatmap.

Reviewers' Comments:

Reviewer #1:

Remarks to the Author:

The authors have address all my previous concerns. However, I still have a minor comment regarding their claim that UAS-Cas9.C is suitable for large-scale genetic screens. They noted that "Importantly, during the screen (see later), lethality was observed in only about 1% of the crosses (despite the use of UAS-Cas9.C), indicating that at least in our experimental system, Cas9.C leakiness-induced lethality is practically a negligible issue." However, the fact that UAS-Cas9.C caused lethality with gRNAs for EcR and Mov34 in a Gal4-independent manner suggests that the Cas9 is non-specifically expressed in tissues critical for metamorphosis. The reason for the low rate of lethality in their ongoing screen is probably because most of the genes they examined are not essential, or at least not required in the leaky-expression tissues for survival. But if someone primarily screens essential genes using UAS-Cas9.C, the lethal rate could be very different. So I suggest the authors to tone down this claim for general screens and acknowledge potential limitations (even though it works well for their particular screen), and even better, to offer practical advice on how to avoid potential problems in screens like this.

Reviewer #2:

Remarks to the Author:

The revised manuscript has addressed my concerns. The clarity in describing the technique, and the availability of resources, has been greatly improved. The authors have also satisfactorily addressed my previous questions regarding quantification of mutant phenotypes.

The work represents a valuable guide and resource for tissue specific screens using CRISPR/Cas9, and will be well-received by the community.

REVIEWERS' COMMENTS:

Reviewer #1 (Remarks to the Author):

The authors have address all my previous concerns. However, I still have a minor comment regarding their claim that UAS-Cas9.C is suitable for large-scale genetic screens. They noted that “Importantly, during the screen (see later), lethality was observed in only about 1% of the crosses (despite the use of UAS-Cas9.C), indicating that at least in our experimental system, Cas9.C leakiness-induced lethality is practically a negligible issue.” However, the fact that UAS-Cas9.C caused lethality with gRNAs for EcR and Mov34 in a Gal4-independent manner suggests that the Cas9 is non-specifically expressed in tissues critical for metamorphosis. The reason for the low rate of lethality in their ongoing screen is probably because most of the genes they examined are not essential, or at least not required in the leaky-expression tissues for survival. But if someone primarily screens essential genes using UAS-Cas9.C, the lethal rate could be very different. So I suggest the authors to tone down this claim for general screens and acknowledge potential limitations (even though it works well for their particular screen), and even better, to offer practical advice on how to avoid potential problems in screens like this.

We thank the reviewer for his positive view of our revisions. We absolutely agree with this comment. We added a reservation that while it worked well for our particular screen we acknowledge that when many of the genes are essential the proportion of lethality may be higher, and that the decision which cas9 to use should be made individually for every screen (one option is to screen with Cas9.C in order to maximize likelihood of positive hits, and switch to Cas9.P2 only for gRNAs that were lethal with Cas9.C)

Reviewer #2 (Remarks to the Author):

The revised manuscript has addressed my concerns. The clarity in describing the technique, and the availability of resources, has been greatly improved. The authors have also satisfactorily addressed my previous questions regarding quantification of mutant phenotypes.

The work represents a valuable guide and resource for tissue specific screens using CRISPR/Cas9, and will be well-received by the community.

We thank the reviewer for his positive view of our revisions.